

# Late Weichselian thermal state at the base of the Scandinavian Ice Sheet

Dmitry Y. Demezhko, Anastasia A. Gornostaeva, Alexander N. Antipin

Institute of Geophysics UB RAS, Yekaterinburg, 620016, Russia

5  *Correspondence to*: Dmitry Yu. Demezhko (ddem54@inbox.ru)

**Abstract.** Geothermal estimates of the ground surface temperatures for the last glacial cycle in Northern Europe has been analyzed. During the Middle and Late Weichselian (55 – 12 kyr BP) a substantial part of this area was covered by the Scandinavian Ice Sheet. The analysis of geothermal data has allowed reconstructing limits of the ice sheet extension and its basal thermal state in the Late Weichselian. Ground surface temperatures outside the ice sheet were extremely low (from -8 to

10  -18 °C). Within the ice sheet, there were both thawed and frozen zones. The revealed temperature pattern is generally consistent with the modern one for the ground surface temperatures in Greenland that makes it possible to consider these ice sheets as analogues. The anomalous climatically induced surface heat flux and orbital insolation of the Earth varied consistently outside the glaciation and independently within the limits of the ice sheet.

## 1 Introduction

During the Weichselian (Valdai) cold stage the Scandinavian Ice Sheet covered the Northern Europe. Despite the fact that the ice sheet has been studied for over a hundred years, its extent, height, the chronology of ice advance and deglaciation, the thermal state at the base of the ice sheet remain the subject of discussion.

The main goal of our study is evaluation of the basal thermal state and extension of the Scandinavian ice sheet in the Late Weichselian (25 – 12 kyr BP) by the analysis of ground surface temperature and heat flux histories, reconstructed from deep-

borehole thermometry data.

Modern ideas about the extension of the Scandinavian ice sheet are very contradictory. Siegert et al. (2001) probed several numerical ice sheet models under different boundary conditions. According to a 'minimum' model the east margin of the Scandinavian Ice Sheet in the Late Weichselian reached the central part of the Kola Peninsula, the border between Karelia and Finland. The ice sheet extended southeastward to the Baltic Sea Gulfs (the Gulf of Finland, the Gulf of Riga and the Gulf of

Gdansk) and westward to the Kattegat covering the Scandinavian Peninsula (Fig. 1). A 'maximal' model implied a much wider extension of the glaciation. A single ice sheet integrated the Scandinavian and the Barents-Kara ice sheets on the east and spread to the British Isles on the west and to the northern Poland and Belarus on the south (Svendsen et al., 2004; Hughes et al., 2016) – Fig. 1.

On the other hand, numerous fossil records (from freshwater clams to mammoths) obtained in central Fennoscandia and dated

back to the Late Weichselian (LW), which were presented at the 31st Nordic Geological Winter Meeting, January 2014, Lund,



Sweden (http://geologiskaforeningen.se/31winter.html) suggest the absence of the single ice sheet. Bolshiyanov (2015) assumes that scattered glacial domes limited in area and height and had a cold frozen base grew in the Scandinavian Peninsula. The evaluation of thermal state at the base of the Scandinavian ice sheet is no less important. Basal thermal state is the main characteristics controlled ice volume and an ice sheet dynamics (Marshall and Clark, 2002; Hughes, 2009) along with glacier

thickness and base relief. The solution of this problem is complicated by the lack of valid evidences of the basal thermal state of the ice sheet. A qualitative estimation is based on the analysis of modern distribution of relict landscapes that mark subglacial thermal regime. Preserved ice-sheet landforms related to frozen-bed conditions, drumlins, flutes and crag-and-tail ridges indicated thawed-bed conditions, and ribbed moraines represent the transition from frozen-bed to thawed-bed conditions (Kleman and Hattestrand, 1999; Kleman and Glasser, 2007).

Direct estimates of the past basal thermal state of the ice sheet can be obtained by the analysis of modern distribution of rock temperatures measured in deep (1.5 – 2 km deep) boreholes (Kukkonen and Safanda, 1996; Kukkonen et al., 1998; 2011; 2015; Mareschal et al., 1999; Demezhko and Shchapov, 2001; Rolandone et al., 2003; Glasnev et al., 2004; Rath and Mottaghy, 2007; Majorowicz and Šafanda, 2008; 2015; Chouinard and Mareschal, 2009; Majorowicz, 2012; Demezhko et al., 2013; 2018; Pickler et al., 2016). However, there are only a few such data all over the world. Those that are located in the region

covered by the Scandinavian ice sheet in the past have been analyzed in the paper.

## 2. Data

The principle of climate reconstruction using borehole temperature data is as follows: if ground surface temperature remains stable then rock's temperature increases with depth with almost constant temperature gradient. Ground surface temperature variations induce temperature anomalies that propagate into the depth and disturb stationary Earth's thermal field. The depth

of temperature anomaly propagation depends on how long ago this anomaly arose on ground surface and on rock's temperature diffusivity value. Analyzing temperature-depth profiles measured in boreholes one can reconstruct the ground surface temperature history and surface heat flux changes.

To date geothermal data were obtained from 11 deep boreholes drilled on the proposed territory of the Scandinavian ice sheet existence (Fig. 1, Tab. 1).

*SG-3.* The results of geothermal investigations of the Kola Superdeep Borehole SG-3 (12261 m deep) were firstly published in the paper (Popov et al., 1999). The first detailed ground surface temperature (GST) history was reconstructed by (Rath and Mottaghy, 2007) using 1-D inversion program based on Tikhonov approach. A limited interval of the temperature-depth profile (500 – 3500 m) devoid of the manifestations of groundwater flow and rock heterogeneities was investigated. To evaluate an optimal regularization parameter GCV method (Farquharson and Oldenburg, 2004) was applied. The GST history for the SG-

3 is shown in Fig. 2.

***Kola (C-1)***. The 2.2 km-deep borehole was drilled in central part of the Kola Peninsula north of the Pana intrusion. GST reconstruction (Fig. 2) revealed extremely low LW temperatures that according to the author points to the absence of a thick ice sheet during the whole Valdai stage (Glaznev et al., 2004).



*Krl.* There are no direct long-term GST reconstructions from this group of boreholes. Seven boreholes with depth from 250 to 700 m were investigated for the heat flow evaluation (Kukkonen et al., 1998). The measured heat flow does not exceed 2.4 – 11.6 mW/m2. It was assumed that such a significant underestimation of heat flow was due to the influence of the last glacial age. To increase the heat flow estimate to a reasonable value of 40 mW/m2 it is necessary to accept that the mean GST 60 – 5   11 kyr BP fall to –15 °C.

*Onega.* According to the GST reconstruction from the 2.5 km-deep Onega parametric borehole the ground surface temperature 25 kyr BP was -14.5 °C, i.e. by 18 – 20 K lower then at present (Demezhko et al., 2013). The increase in surface temperature began quite early (about 20 kyr BP) and as proposed was associated with the insulating effect of the Scandinavian ice sheet. Only after the ice sheet termination about 12 kyr BP the GST variations began to reproduce a global warming.

10  *Outokumpu.* Two efforts were made to reconstruct the GST history using temperature-depth profile logged in the 2.5-km-deep Outokumpu borehole. Firstly, the reconstruction was obtained by solving the direct heat conduction problem for the standard model of temperature changes in Finland using adjustment of the past temperature variation amplitudes (Kukkonen and Safanda, 1996). The evaluated mean GST over a period from 100 to 10 kyr BP was equal to -2°C. Second time, the GST history was reconstructed in more detail (Kukkonen et al., 2011) using the inversion program based on Tikhonov approach 15  (Rath and Mottaghy, 2007). In this case, the ground surface temperature in the period from 30 to 40 kyr BP proved to be slightly lower, i.e. -4°C.

*Olkiluoto.* The official title of the borehole is OL-KR56. It was drilled in 2012 to a depth of 1200 m on Olkiluoto Island in the Gulf of Bothnia (Toropainen, 2012). Several inversion approaches were applied to reconstruct the GST history (Kukkonen et al., 2015) including SVD (singular value decomposition), deterministic inversion approach, i.e. Tikhonov technique (Rath and 20  Mottaghy, 2007) and method based on Bayesian stochastic approach (Markov Chain Monte Carlo technique). The median temperature history for the ensemble of reconstructions is presented in Fig. 2. A very slight increase of temperature since the end of the glacial age is probably due to the fact that the region under study was flooded after the glacial termination up to the historical time (1 kyr BP) (Smellie et al., 2014).

*Gravberg-1.* The Gravberg-1 Superdeep Borehole (6337 m-deep) is located on the north of the Siljan impact structure in central 25  Sweden. No direct reconstructions of GST histories were made here. Geothermal investigations did not reveal any significant variations of geothermal gradient down to the depth of 5 km (Balling et al., 1990) that indicates the absence of significant changes of ground surface temperature during the last glacial cycle. The results of PMIP (Palaeoclimate Modelling Intercomparison Project's) modeling of the dynamics of the ice sheet proved that LW ground surface temperature did not increase above -2 – -4°C (Forsström, 2005).

30  *Ullrigg.* We used the borehole data published in (Maystrenko et al., 2015) to reconstruct the GST history for the Ullrigg borehole located in western Norway near the North Sea coast. The inversion algorithm used in this study (Demezhko and Shchapov, 2001) allows reconstructing GST history as a step function with uneven time intervals: the duration of the intervals increase into the past. The set of GST histories with a different number of steps equally satisfying the measured temperature

log was averaged and smoothed. The final reconstruction was adjusted for the insufficient restoration of thermal equilibrium after the termination of drilling. After that correction the LW temperature fall from +4.5°C to +1.4°C remaining however positive. As in case of Olkiluoto a small amplitude of past temperature variations may reflect the fact that the region was flooded most of the time after the glacial termination.

*Forsmark.* Eight boreholes located along the Baltic Sea coast in eastern Sweden were drilled to a depth of 800-1000 m into a metagranite and measured in 2003-2006. To reconstruct paleoclimate all temperature-depth profiles were merged and combined with averaged rock properties. Various inversion methods and approaches including deterministic inversion approach (Tikhonov techniques) and Bayesian stochastic approach (Markov Chain Monte Carlo technique) were used (Rath et al., - in press). We analyzed GST reconstructions obtained by the deterministic inversion approach. 20 – 30 kyr ago the GST

in the region under study was equal to -1°C.

*Laxemar.* The borehole with a depth of about 1450 m is also located on the Baltic Sea coast of Sweden in 340 km to the south of Forsmark. Temperature logging was conducted in 2003. As in the previous case, we analyzed GST reconstructions obtained by the deterministic inversion approach (Rath et al., - in press). The LW temperature was about 0 °C here.

*Udrin.* The Udrin borehole is situated in northeast Poland within the limits of anorthositic intrusion concealed under 800-m

thick sedimentary layer.  Negative geothermal gradient to a depth of 400 m was explained by long melting of a thick permafrost formed under the extremely low temperatures of the last glaciation (Šafanda et al., 2004). Solving the direct heat conduction problem taking into account phase transition it was found that the mean temperature of the last glacial age was equal to -10°C. The inversion has showed that temperature in the last glacial cycle did not fall below -8°C (Rath and Mottaghy, 2007).

### 3. Spatial pattern of the Scandinavian ice sheet basal temperatures in the Late Weichselian (Valdai)

The Scandinavian ice sheet (Fig.1) demonstrates a large variety of GST in the Late Weichselian. One can to divide all GST estimates into three groups. The first group obtained on the eastern and southern margins of the region (central part of the Kola Peninsula, Karelia and the northern Poland – boreholes Kola, Krl, Onega, Udrin) demonstrates an extremely low LW temperatures from -8 to -18°C. Such temperature regime at the ground surface points to the existence of ice-free conditions during much of the Late Weichselian. At least if the glacier existed here then it was not for a long time and its thickness was

not so significant to make any noticeable contribution in the modern thermal field. It is clear that a thick permafrost formed here under such a low annual temperatures.

The second group of GST estimates (the southern Norway, Sweden and Finland – boreholes Ullrigg, Forsmark, Laxemar, Olkiluoto) characterizes the warmest conditions at ground surface in the Late Weichselian with GST from -1 to +2°C. These temperatures exceeded an ice melting point considering the effect of glacier pressure. The presence of melt water at the base

of the glacier ensured the possibility of its horizontal movements.



The third group of LW temperature estimates (boreholes SG-3, Outokumpu, Gravberg) represents the medium conditions (-3 – -4°C) which is typical for a frozen base of the ice sheet. These few GST estimates are limited to a central part of the studied region and probably to ice divides of the Scandinavian ice sheet.

For pure ice the melting point at the base of a glacier $T_m$ depends on the pressure (Aschwanden, 2014):

$$T_m = T_t - \gamma(\rho g h - p_t),\tag{1}$$

where $T_t = 273.16$ K and $p_t = 611.73$ Pa are the triple point temperature and pressure of water, $\rho = 900$ kg/m³ is an ice density, $g = 9.825$ m/s² is acceleration due to gravity, $h$ is a glacier thickness, $\gamma = 7.42 \cdot 10^{-8}$ K/Pa. For the including of air bubbles, the value of $\gamma$ may be up to $9.8 \cdot 10^{-8}$ K/Pa (Harrison, 1975) which makes the dependence of melting point on pressure steeper. The presence of NaCl and KCl shifts the melting point towards negative temperatures by the value of $0.064 C$ where $C$ is the salt content in g/l (Galushkin, 1997). The latter ratio is largely relating not to the glacier that is sufficiently pure with respect to salts but to the ice contained in rocks of the glacial base.

The differences in the thermal state of individual zones are most clearly seen when comparing the GST reconstructions with the data of an empirical model of spatial distribution of Pleistocene/Holocene warming amplitudes summarizing a number of long-term geothermal reconstructions obtained earlier in North Eurasia (Demezhko et al., 2007). Generally, these GST estimates were obtained outside the studied region (Fig. 3). According to the model, the amplitudes of Pleistocene/Holocene warming $\Delta T$ decrease inversely with the distance $r$ (km) between the center of warming located in the North Atlantic ($\varphi_0 = 71,12°$ N, $\lambda_0 = 0,31°$ W) and observation point with coordinates $\varphi$, $\lambda$:

$$\Delta T(r) = 1.75 + 2.714 * 10^{-4} r^{-1}, \quad r = r(\varphi_0, \lambda_0, \varphi, \lambda).\tag{2}$$

Subtracting the modeling amplitudes of Pleistocene/Holocene warming from the current mean annual ground surface temperatures $T_c$ at the borehole location points, we obtain rough estimates of the 'normal' GST $T^m_{LW,}$ which characterize the temperature regime while there was no ice sheet (Tab. 2). In Fig. 4 these 'normal' modeling temperatures are comparing with the reconstructed ones.

GST estimates obtained outside the ice sheet (boreholes Kola, Krl, Onega, Udrin) are in a good agreement with the model (2). On the contrary GST estimates from boreholes SG-3, Outokumpu, Gravberg, Olkiluoto, Ullrigg form a compact cluster $T_{LW} >> T^m_{LW}$ representing the insulating effect from the ice sheet.

## 4. Distribution of the ground surface heat fluxes

Besides the ground surface temperature and its variations in a glacial cycle an important characteristic of the thermal regime is change in the anomalous surface heat flux (SHF) caused by climate change (Demezhko et al., 2013; Demezhko, Gornostaeva, 2014; 2015 a,b).

It does not do to confuse the anomalous surface heat flux (SHF) with geothermal heat flow. The latter is considered to be stationary and associated with the internal sources of heat. Geothermal heat flow is positive in an upward direction (from depth





to ground surface). SHF is a non-stationary component of the total radiation at ground surface associated with the variations of incoming and outgoing heat fluxes. SHF is positive in downward direction (from the surface to depth).

Being an energy expression of climate variations this heat flux contrary to temperature might be directly compared with another energy characteristics (for example with solar insolation) to identify the origin of climate changes. The algorithm of GST –

SHF transformation have been developed for SHF evaluation (Demezhko, Gornostaeva, 2015a). SHF histories obtained from GST histories mentioned above are shown in Fig. 5. SHF histories obtained outside the ice sheet (boreholes Kola, Onega, Udrin) demonstrate a positive SHF anomaly from 20 to 5 kyr BP that correlate with the insolation anomaly at 60°N associated with the variations of the Earth orbital parameters (Berge and Loutre, 1991). SHF anomalies for the Kola and Onega boreholes are more pronounced than the same for the Udrin borehole although they are complicated by the local minima. Coincident

variations of the SHF and insolation are typical for the regions far from the Pleistocene glaciation area, for example for the Middle Urals (Demezhko, Gornostaeva, 2015a,b). However, we observed such synchronicity on the territory of the Laurentide ice sheet too (Demezhko et al., 2018). In (Demezhko, Gornostaeva, 2015a,b) the ratio of the SHF to the insolation change was proposed to be considered as the Earth's climate sensitivity. This parameter determines how much of the additional energy incoming to the upper boundary of the atmosphere due to the variations of the Earth's orbital parameters was finally spent to

change the ground surface temperature.

The climate sensitivity value for Kola and Onega boreholes amounts to 1.2% and 1.5% respectively. For the Udrin borehole the climate sensitivity value is significantly lower (only 0.6%). It seems to be that here most of the additional heat flux from the insolation increase was spent to melting of the ice contained in high-porous frozen sedimentary rocks. It is clear that latent heat fluxes spent on phase transitions cannot be reconstructed by the GST-SHF transformation.

Only one SHF reconstruction (from the Outokumpu borehole) within the limits of the Scandinavian ice sheet demonstrates the SHF variations similar to the insolation ones. The climate sensitivity value for this region amounts to 0.6%. By comparison, the climate sensitivity was estimated at 1.3–1.5% in the Middle Urals (Demezhko, Gornostaeva, 2015a,b) and at about 1% in Canadian province of Alberta under the ice sheet (Demezhko et al., 2018).

Thus in the regions which are not covered by the ice sheet the SHF and consequently the GST changes are caused by the

insolation variation. The glacier either diminishes solar influence or completely eliminates it.

## 5 Discussion

Unexpected result of the investigation is an extremely low LW temperatures on the eastern and southern margins of the studied region which were from -8 to -18°C that is lower than current mean annual ground surface temperatures by 15–20°C. In the process of modeling of the Scandinavian ice sheet it is usually assumed that LW temperatures has dropped at the most by 5–

10 °C lower than modern temperatures (Siegert et al., 2001). At such low temperatures, different cryogenic structures were formed in this region. Evidence of such structures could be preserved in the present.



## 5.1 Permafrost features

Evidences of paleopermafrost are ice-wedge pseudomorphs, frost-cracking structures, thermal contraction cracks, cryoturbations in sandy, sandy-clay sediments and peats (Harry and Gozdzik, 1988; Huijzer and Vandenberghe, 1998; van Huissteden et al., 2003). Few such features were found on the territory of Sweden, Norway and Finland covered by the ice sheet during most of the Late Weichselian. In Suupohja region in western Finland permafrost features are over 65 kyr old. They arose before the late Weichselian glaciation (Pitkäranta, 2009). In the southwestern Sweden and the northernmost Norway, the ice-wedge casts findings are probably younger and associated with the early deglaciation of these regions (Svensson, 1988). Thin permafrost in the eastern Finland was formed only in Younger Dryas (from 12.9 to 11.7 kyr BP) (Donner et al., 1968). It is clear that such a short period could not leave a thermal trace that has preserved to the present. Most of permafrost features were found on the North of Germany, Poland and Denmark (see overview in van Huissteden et al., 2003). The ice-wedge casts are dated to the period from 27 to 47 kyr BP. Under general climate cooling and the existed ice-free conditions in the Late Weichselian thick permafrost was formed here which was reflected in geothermal reconstruction from the Udrin borehole (Šafanda et al., 2004; Rath and Mottaghy, 2007).

One can expect a lot of paleopermafrost features in eastern part of studied region (in Karelia and the east part of the Kola Peninsula) where an extremely low LW temperatures were reconstructed. However, sandy-clay sediments and peats occurring here are not older then the Holocene because an olden sediments were eliminated by the last ice sheet. Few ice-wedged casts were found near the eastern coast of Onega Lake (Demidov, 2006) and on the southern coast of the Gulf of Finland (Rusakov, Nikonov, 2010; Streletskaya, 2017; Shvarev et al., 2018). The researcher's opinions regarding the origin of the latter findings are different. Thus, Streletskaya (2017) consider that wedge structures are the evidence of permafrost formed after 15.5 kyr ago. Mean annual ground temperature reconstructed using the cryogenic contrast ratio were lower than -8 ºC. Other researchers (Rusakov, Nikonov, 2010; Shvarev et al., 2018) suppose that wedge structures are associated with seismic activity.

## 5.2 Spatial pattern of LW temperatures and modern seismicity

The region with extremely low LW temperatures coincides with the area in which modern seismic activity is practically absent (Fig. 6). It is likely not a spurious coincidence. Both fields namely paleotemperature and seismic ones demonstrate effects accumulated over a long time.

The inversion of borehole temperature data allows estimating the ground surface temperature not at specific time $t$ in the past but the mean temperature for the period $t \pm t/3$ (Demezhko and Shchapov, 2001). For example, temperature 21 kyr BP on the reconstructed GST history represents an average value over a period of 14 – 28 kyr BP. Therefor we cannot to state definitely (like Glaznev et al., 2004 did) that GST estimates confirm the 'minimum' model of the Scandinavian ice sheet in the Late Weichselian. It is possible that thermal trace of glaciation on the eastern and southern margins of the region did not remain because of negligible duration of the ice sheet existence here and/or its small thickness.



Contemporary seismic activity in the region reflects the influence of two factors such as isostatic adjustments following the melting of the ice sheet and tectonics that is not associated with glaciation. Morner (1979) separated two components in postglacial uplift, one is linear and another is exponential. The linear component has a tectonic origin, the exponential one is associated with isostatic adjustments. At that, the exponential component constitutes a small part of the total uplift. Fjeldskaar

et al. (2000) hold an alternative view. Comparing the measured rates of uplift with the results of the ice sheet modeling, they conclude that isostatic adjustments are the main origin. Maximal uplift rates and maximal seismic activity are in the regions covered by thick ice sheet during most of the Late Weichselian. The absence of any seismic activity on the east and the south of the studied region may indicates a short duration of the glacier existence here.

### 5.3 Comparison with the basal thermal state of the Greenland ice sheet

Spatial pattern of the LW ground surface temperatures at the base of the Scandinavian ice sheet is in general consistent with the present GST zonation in Greenland. The boreholes drilled here (Fig. 7) revealed a frozen bed in central part of the glacier near the ice divide (Camp Century, DYE-3, GISP2, GRIP, NEEM).

The detailed distribution of frozen and thawed zones at the base of the Greenland ice sheet was recently reconstructed by MacGregor et al. (2016). The reconstruction is based on the data from remote sensing with spectral radiometer MODIS, the

data on the present day glacial movements, and the mathematical simulation (Fig. 7). A comparatively narrow zone of frozen rocks at the base of the glacier spreads from the north to the south. Extensive thawed zones are located in peripheral parts of the glacier, i.e. east and west of the frozen zone. Outside of the ice sheet especially in northern Greenland mean annual ground surface temperatures (according to weather station data – Schomacker et al., 2017; Xing, 2014) are significantly below zero, i.e. -6 ºC in Pituffik, -8.4 °C in Danmarkshavn, -12.1 °C in Bliss Bugt.

An existence of thawed zones in the base of the glacier is often considered as an indication of its intensive thawing and rapid termination (Leeson et al., 2015). There are different opinions about the causes of thawed zones occurrence. Basal temperature is usually related to the value of geothermal heat flow from the Earth's interior (Pickler et al., 2016; MacGregor et al., 2016; Rogozhina et al., 2016). Certainly geothermal heat flow has a part in the formation of glacier thermal state however not crucial. The simulation results (Demezhko et al., 2007) showed that temperature at the base of the glacier depends on the balance of

heat flow, vertical advection and a glacier height influences. Under significant glacier height and high vertical advection rate low temperatures from a glacier surface are transmitted to its basement. Convective heat transfer mechanism is more effective than conductive one. As a result, the glacial basement may cools. The validity of this viewpoint is verified by the high concentration of frozen zones near the ice divide of the Greenland ice sheet where its height and a vertical component of ice movement rate are maximal.

30   A. Leeson with coauthors (Leeson et al., 2015) relate the formation of thawed zones to so-called supraglacial lakes. Forming in warm half year on a glacier surface these lakes effectively absorb solar energy and then penetrate to the glacial basement increasing the intensity of its melting and horizontal movements.  However even at present period of global warming these



processes cover only a narrow 100-km zone along the Greenland ice sheet margins that represents very small part of its thawed zone.

The similarity of the spatial pattern of thawed and frozen zones at the base of the Greenland and Scandinavian ice sheets allows considering them as close analogies. Since the warm base of the Scandinavian ice sheet according to reconstructions preserved

for several tens of thousands of years there is no reasons to believe that the Greenland ice sheet is now unstable.

## 6 Conclusion

Late Weichselian GSTs estimated from borehole data in Northern Europe demonstrate a high variability from -18°C to +2°C. The first group of GST estimates obtained on the eastern and southern margins of Fennoscandia (in central part of the Kola Peninsula, in Karelia and northern Poland) detects extremely low ground surface temperatures from -8 to -18 °C. Such thermal

state of the ground surface indicates the absence of an ice sheet here during the most part of the Late Weichselian. At least, if the glacier existed here then it was not long (less than 10 ka) and it was not high enough to induce an anomaly that could persist to the present. Thick permafrost was formed in these regions under such a low mean annual temperatures. Evidences of past cryogenic processes must be currently presented here. The second group of GST estimates (the south of Norway, Sweden and Finland) displays the warmest conditions at the ground surface in the Late Weichselian with temperatures from -

1 to +2 °C. These temperatures exceeded the ice melting point considering the effect of glacier pressure that probably determined a significant horizontal dynamics of the Scandinavian ice sheet. The third group of GST reconstructions (the western part of the Kola Peninsula, central Finland, and central Sweden) represents the medium temperature conditions (from -3 to -4°C) which is typical for a frozen base of the ice sheet. These estimates of LW ground surface temperature are limited to a central part of the studied region and probably to ice divides of the Scandinavian ice sheet.

The area with the extremely low reconstructed LW temperatures is located in aseismic region. While on the territory covered by the Scandinavian ice sheet much of the Late Weichselian a high seismic activity is now observed that is probably due to the isostatic adjustments after the glacier termination.

Spatial pattern of LW ground surface temperatures at the base of the Scandinavian ice sheet is in general consistent with the present GST zonation at the base of the Greenland ice sheet that makes it possible to consider them as close analogies. Since

thawed zones at the base of the Late Pleistocene ice sheets remained several tens of thousands of years there is no reasons to believe that such zones at the Greenland ice sheet basement are the feature of its instability and degradation.

The ground surface heat flux changes outside the Scandinavian ice sheet occurred coincidently with the variation of the Earth's insolation. On the contrary, SHF and insolation varied independently within the limits of the glaciation. The exception is the data from central Finland (the Outokumpu borehole) where solar impact is probably implemented through changes of

30 the thermomechanical properties of the glacier. We observed such coincidence of SHF and insolation variations on the territory of the Laurentide ice sheet (the Hunt Well – Demezhko et al., 2018). However further studies are needed to investigate the mechanism of such coincidence under the ice sheet.



**Author contributions**

DYD initiated and supervised this study. DYD and AAG conducted the investigation. AAG and ANA designed the algorithm and performed GST to SHF transformation. All authors contributed to the discussion and interpretations.

**Competing interests**

The authors declare that they have no conflict of interest.

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

**Table 1 The reconstructed GST in the Late Weichselien ($T_{LW}$), the estimates of the ice sheet height ($h$) according to the 'minimal'**
10 **(Siegert et al., 2001) and 'maximal' (Svendsen et al., 2004) models and the ice melting point estimates ($T_m$) calculated by Eq. (1) for pure ice.**

| № | Borehole | Location | $T_{LW}$, °C | $h$, m | | $T_m$, °C | | Reference to the GST reconstruction source |
|---|----------|----------|--------------|--------|--|-----------|--|--------------------------------------------|
| | | | | min | max | min | max | |
| 1 | SG-3 | 69°23' N 30°36' E | -3 | 750 | 2150 | -0.5 | -1.4 | Rath and Mottaghy, 2007 |
| 2 | Kola (C-1) | 67°45' N 35°25' E | -18 | | 2350 | | -1.5 | Glaznev et al., 2004 |
| 3 | Krl | 63°15' N 36°10' E | -15 | | 700 | | -0.4 | Kukkonen et al., 1998 |
| 4 | Onega | 62°09' N 34°24' E | -14.5 | | | | | Demezhko et al., 2013 |
| 5 | Outokumpu | 62°43' N 29°04' E | -4 | 1000 | 2600 | -0.6 | -1.7 | Kukkonen et al., 2011 |
| 6 | Olkiluoto | 61°14' N 21°30' E | +2 | 2000 | 2500 | -1.3 | -1.6 | Kukkonen et al., 2015 |
| 7 | Gravberg | 61°09' N 15°00' E | -3 | 2300 | 2600 | -1.5 | -1.7 | Forsström, 2005 |
| 8 | Ullrigg | 58°56' N 5°42' E | 1.4 | 100 | 2250 | -0.1 | -1.5 | GST reconstructions from the data by (Maystrenko et al., 2015) |
| 9 | Forsmark | 60°23' N 18°12' E | -1 | | | | | Rath et al., - in press |
| 10 | Laxemar | 57°25' N 16°38' E | 0 | | | | | Rath et al., - in press |
| 11 | Udrin | 54°14' N 22°56' E | -8 | | 100 | | -0.1 | Rath and Mottaghy, 2007 |





**Table 2 The Late Weichselian and current GSTs: $T_{LW}$ is the reconstructed GST; $T_C$ is the current annual GST; $\Delta T$ is the amplitude of Pleistocene/Holocene warming according to the model (Demezhko et al., 2007), $T^m_{LW} = T_c - \Delta T$ is the modeling 'normal' LW temperatures.**

| № | Borehole | $T_{LW}$, °C | $T_c$, °C | Modeling estimates | | References |
|---|----------|------|------|----------|----------------------|-----------|
| | | | | $\Delta T$, °C | $T^m_{LW} = T_c - \Delta T$, °C | |
| 1 | SG-3 | -3 | 1.5* | 24 | -22.5 | Rath and Mottaghy, 2007 *Xing, 2014 |
| 2 | Kola (C-1) | -18 | 2.0 | 21 | -19 | Glaznev et al., 2004 |
| 3 | Krl | -15 | 4.3* | 17.5 | -13.2 | Kukkonen et al., 1998 *Xing, 2014 |
| 4 | Onega | -14.5 | 5.5 | 17.5 | -12 | Demezhko et al., 2013 |
| 5 | Outokumpu | -4 | 5 | 22 | -17 | Kukkonen et al., 2011 |
| 6 | Olkiluoto | +2 | 5.5* | 21.5 | -16 | Kukkonen et al., 2015 *Luterbacher et al., 2004 |
| 7 | Gravberg-1 (Siljan) | -3 | 6* | 23 | -17 | Forsström, 2005 *Xing, 2014 |
| 8 | Ullrigg | 1.4 | 5.6* | 22 | -16.4 | $T_{LW}$ reconstructions from the data by (Maystrenko et al., 2015). *Xing, 2014 |
| 9 | Forsmark | -1 | 6.5* | 20.5 | -14 | Rath et al., - in press *Xing, 2014 |
| 10 | Laxemar | 0 | 7.3* | 17.7 | -10.4 | Rath et al., - in press *Xing, 2014 |
| 11 | Udrin | -8 | 7 | 14.1 | -7.1 | Rath and Mottaghy, 2007 |

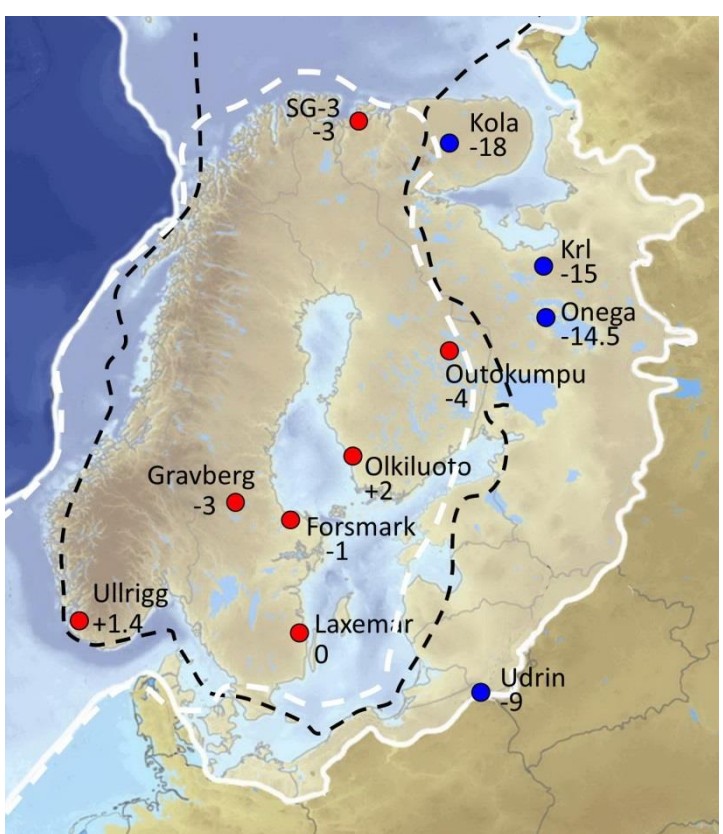

**Figure 1: Late Weichselian (LW) Scandinavian ice sheet extension and reconstructed LW surface temperatures. White solid line indicates maximum ice-margin extension 20 kyr BP (Hughes et al., 2016), white dashed line indicates ice-margin position 27 kyr BP (Hughes et al., 2016), black dashed line – LW ice extension according to minimum model of Siegert et al. (2001). Color circles mark**
5 **the explored boreholes, numerical labels – the reconstructed LW ground surface temperatures (see also Tab. 1).**




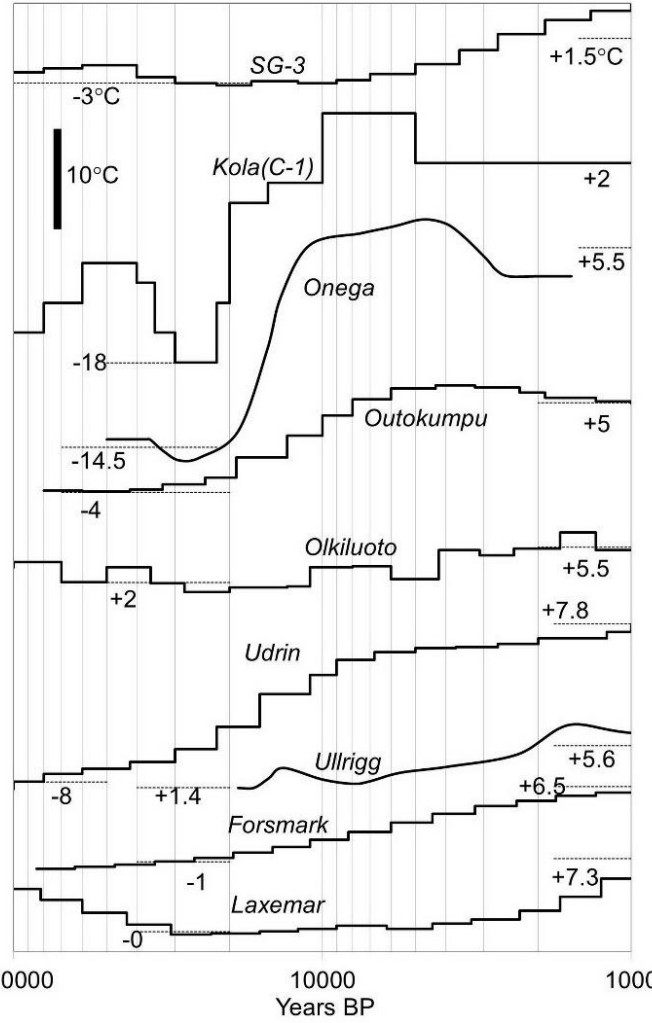

**Figure 2: GST reconstructions. Dashed line and numerical lables mark LW temperatures in the glacial maximum and modern ground surface temperatures (see Tab. 1, 2). The shown GST histories begin from 1 kyr BP.**





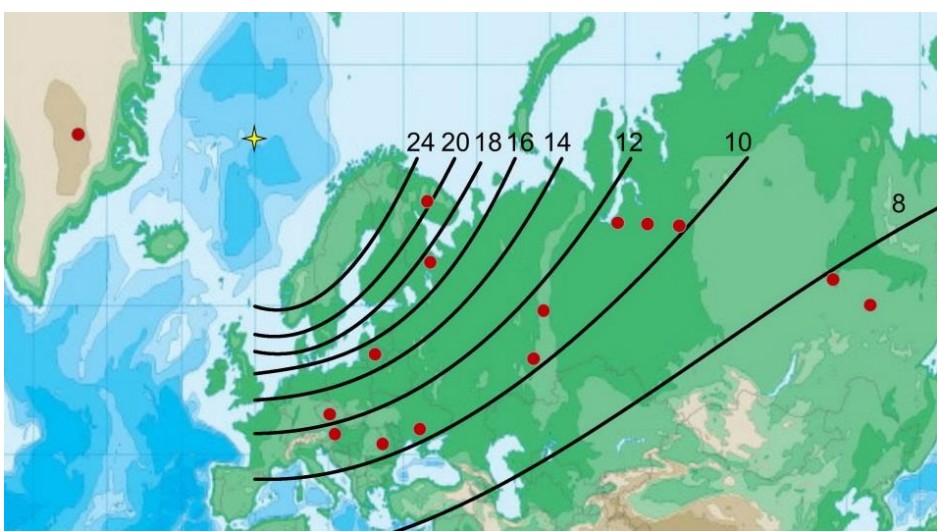

**Figure 3: Spatial distribution of the amplitudes of Pleistocene/Holocene warming ΔT in North Eurasia according to the model (2). Red points mark the boreholes for which long-term GST reconstructions have been obtained. Yellow star is the center of warming (Demezhko et al., 2007).**





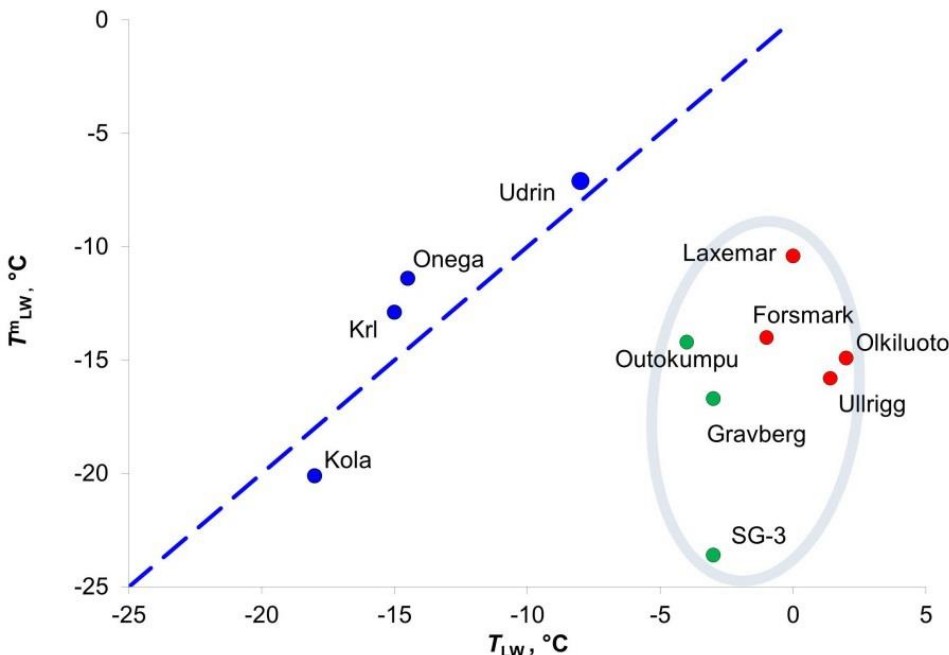

**Fig. 4. Comparison of the reconstructed LW surface temperatures $T_{LW}$ with the model 'normal' temperatures $T^m_{LW}$ characterizing thermal regime in the absence of the ice sheet. Dashed line corresponds to the ratio $T_{LW} = T^m_{LW}$.**



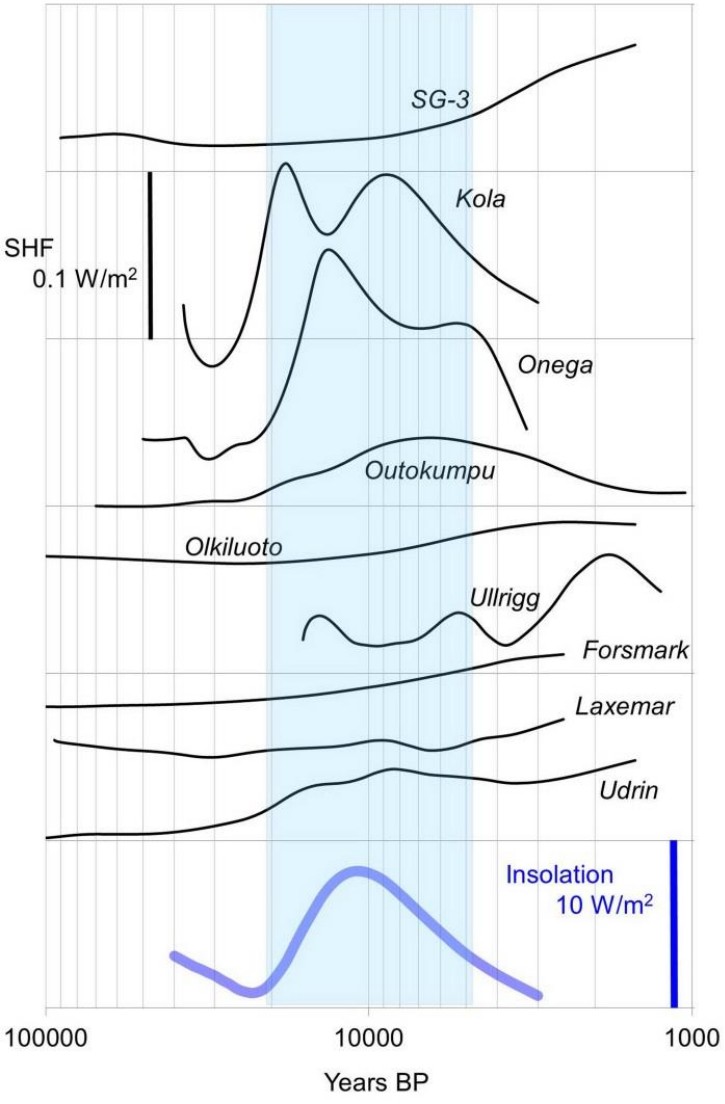

**Figure 5: SHF histories (thin solid lines) and insolation changes at 60° N (heavy blue line – Berge and Loutre, 1991). The thermal effusivity value was universally taken equal to $E = 2.5 \cdot 10^3$ J·m$^{-2}$·K$^{-1}$·s$^{-1/2}$ for the SHF reconstruction. The insolation curve was preliminary smoothed by the procedure presented in (Demezhko and Gornostaeva, 2015b) for coordination with time resolution of geothermal reconstructions.**



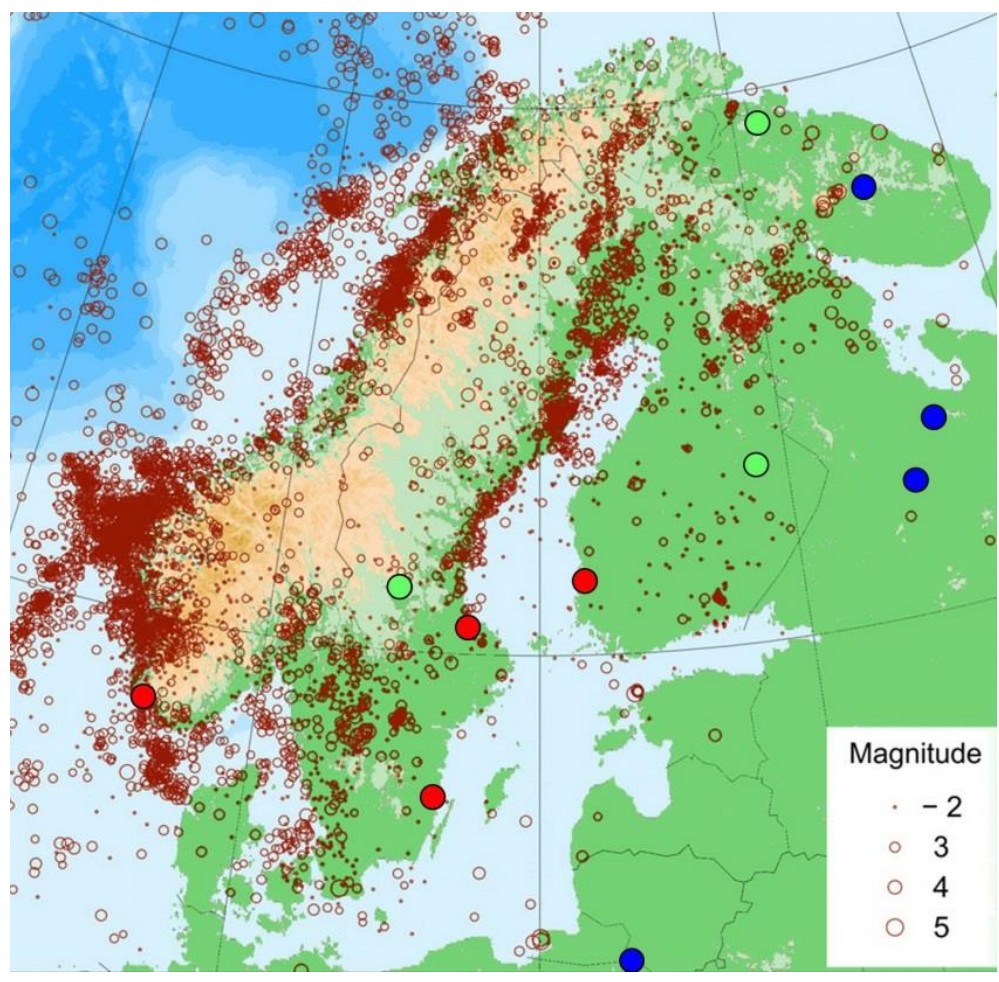

**Figure 6: Distribution of the earthquake epicenters in northern Europe (1971- 2012) according to the Institute of Seismology of the University of Helsinki: http://www.seismo.helsinki.fi/bulletin/list/catalog/instrumap.html. Color circles mark the explored boreholes: blue– sites, where LW ground surface temperatures were extremely low (-8 – -18 ºC); green – sites of frozen bed; red – sites of thawed bed.**





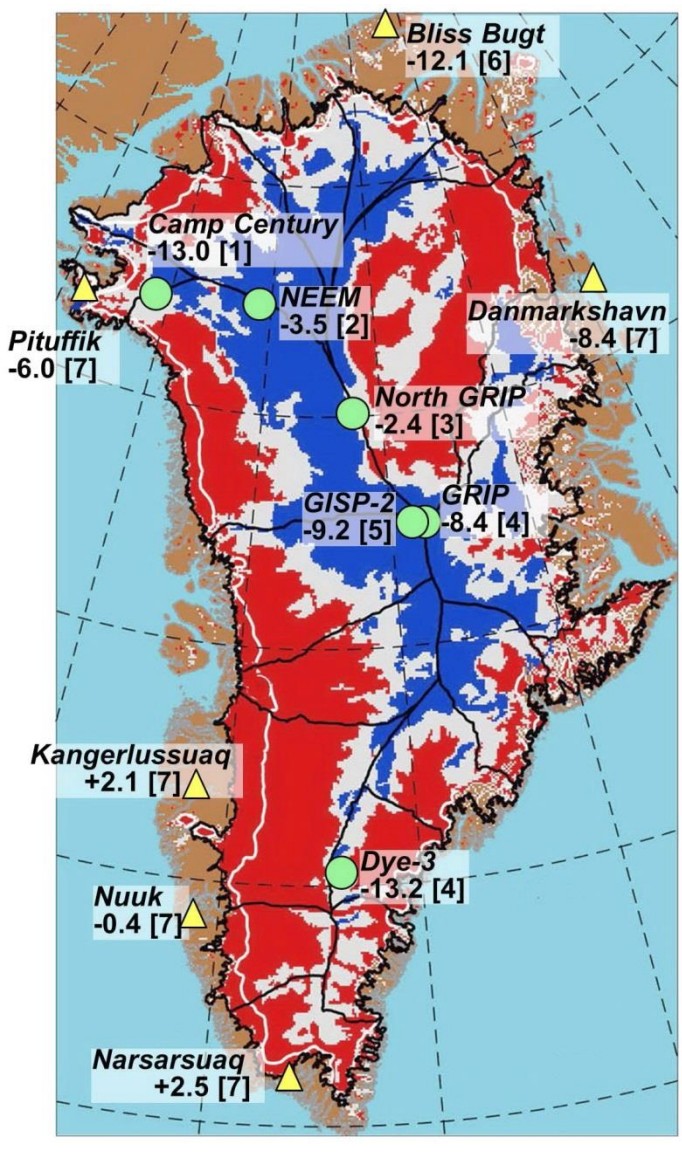

Figure 7: The basal thermal state of the Greenland ice sheet. Areas with thawed bed are shown in red, with frozen bed – in blue, with uncertain thermal state – in grey (MacGregor et al., 2016). Green circles mark the boreholes drilled in the Greenland ice sheet, numerical labels denote basal temperature and citation (in squire brackets: 1 – Weertman, 1968; 2 – MacGregor et al., 2016; 3 –
5  Dahl-Jensen et al., 2003; 4 – Dahl-Jensen et al., 1998; 5 – Cuffey et al., 1995). Triangles are weather stations located out of the ice sheet, numerical labels under them – mean annual GST and citation (in squire brackets:  6 – Schomacker et al., 2017; 7 – Xing, 2014).