# Peer review of "Late Weichselian thermal state at the base of the Scandinavian Ice Sheet"

_Climate of the Past, 2019_

## Referee Comment (RC1) · Anonymous Referee #1 · 29 Jun 2019

It is an excellent paper giving new interpretation of the variability of TLW temperatures in relationship to glacial ice regional reach in time and space.

There are just minor things to be fixed:

1. It is Udryn not Udrin.

2. I propose to give a range of TLW in Table 2 for Udryn including earlier modelling result TLW=-10C of Safanda et al. 2004 (cited in the references) versus existing much later estimate by Rath and Mottaghy (2007), (-8C). I would give a range -8 to -10C as shown in both publications which used different methodology.

---

## Referee Comment (RC2) · Anonymous Referee #2 · 7 Jul 2019

The main goal of this manuscript by Demezhko et al., as clearly stated in the introduction, is an evaluation of the basal thermal state and extension of the Scandinavian ice sheet in the Late Weichselian (25 – 12 kyr BP) using ground surface temperatures and heat flux histories as derived from deep-borehole geothermal data. Data from 11 boreholes from the region of study are applied.

The core results consist of a compilation of published/reported results with the extraction of Late Weichselian (LW) ground surface temperature (GST), either from original author's model results or supplemented with new modelling results by the authors. Unfortunately, it is not always clear to the reader, when new modelling results are included.

The main idea of the study is very good, however, the borehole data and resulting temperature histories are mostly of insufficient quality for the conclusions. The extraction

of long-term past GST history from deep-borehole geothermal data is possible only if high-quality temperature-depth data are applied (undisturbed by the drilling process, ground-water movement etc.) and from boreholes of sufficient depth and sufficient information on rock thermal conductivity and heat production. Apparently, so far, only rather few such good results are available, such as that of Kukkonen et al. (2011; Physics of the Earth and Planetary Interiors, 188, 9–25) and Dahl-Jensen et al. (1998; Science, 282, 268–271) from a different environment - the Greenland ice sheet.

In addition to using data of generally insufficient quality, there is not much information on the methodology applied and no discussion on uncertainties of the extracted main temperature parameter, the Late Weichselian ground surface temperature. As an illustration of the apparent lack of sufficient emphasis (and understanding) of potential uncertainty, the authors indicate an uncertainly of the time of an extracted level of GST as t +- t/3. This is illustrate with the example of a t-value of 21 kyr BP where the "reconstructed GST history represents an average over a period of 14 – 28 kyr BP" (p. 7). This is clearly too simple. There is also uncertainty on the amplitude of temperature variations and a tradeoff between time and amplitude.

Still, the main problem here is the quality of the applied GST histories. The data from Kukkonen et al. (2011) from the Outokumpu deep borehole in SE Finland, mentioned above, are used and seem of good quality. Most other results are of far less quality. This is apparent from Fig. 2, which shows the applied GST reconstructions.

Starting from the top, with SG-3, we see an almost linear trend of increasing temperature (from c. -3°C to positive) from between 10 and 5 kyr BP and up to 1 kyr BP (present GST is +1.5°C). Similar unrealistic long trends of temperature increase up to recent times are seen also in the data from Forsmark, Laxemar and Ullrik. These 'reconstructions' are clearly inconsistent with the general knowledge of past climate in these areas as well as inconsistent with the applied data from boreholes of better quality.

Among the sites with very low LW GST estimates are the Kola (C-1), Krl and Onega boreholes. For Onega, we see an 'unexpected' drop in temperatures by more than 5°C from c. 5 to 2.5 kyr BP. Looking into details of original borehole temperature data (in Demezhko et al. 2013), we observe too high near surface temperatures (c. 11°C significantly above present day GST of c. 5.5°C). The applied borehole temperature data are clearly disturbed by the drilling process. A correction is attempted resulting in an unrealistic 'warming period' and the above 'unrealistic' temperature drop. This results in too large amplitude of the temperature rise from c. 20 to 10 kyr BP and significant uncertainty on the Late Weichselian temperature estimate of - 14.5°C.

For the Kola (C-1) site, a LW GST estimate of -18°C is indicated. The problem here is a 'deep narrow cooling' between c. 35 and 20 kyr BP. Such a narrow time interval of low temperature is very unlikely to be resolved from borehole data and imposes significant uncertainty on the GST estimate. For borehole Krl, no GST history is given, and the low LW value of -15°C is obtain by 'selecting' an estimated 'unperturbed' heat flow of 40 mW/m2 without any mention of modelling procedure, nor information on deep background heat flow (why 40?). Again, significant uncertainty. It is likely, that this last group of boreholes may show quite low LW GST estimates, but there is a lack of critical evaluation and no discussion of uncertainty levels.

Without a more detailed analysis of original borehole geothermal information, it is difficult to point to general or specific reasons for obtaining often unrealistic GST histories or GST histories with great uncertainty. As indicated above, main sources include insufficient depth of borehole temperature data, lack of representative information on rock thermal properties (variability of conductivity) and temperature disturbances from ground water migration. In addition, modelling procedures often seem to underestimate uncertainty limits and the possibility of applying independent prior information.

There are also other aspects of this manuscript, which point in a clearly negative direction. Two examples are given:

[Figure]

The 'mathematical/numerical' contouring of data in Fig. 3 and the extrapolation into the Scandinavian region (with "a center of warming" located in the North Atlantic) has so large uncertainties in the region of study, that it should not be used for a detailed treatment as in Table 2 and Fig. 4 and associated discussion.

The notion of a potential correlation between the region of very low modern seismicity (Fig. 6) and very low LW temperatures seems highly speculative. Most of Finland has very little seismicity, also in areas of significant ice thickness towards Gulf of Bothnia. The highest current seismicity is in southwester Norway in areas along the ice sheet margin.

Among the positive elements of this study is the compiled data of "basal thermal state" of the Greenland ice sheet (Fig. 7), shown for a comparison and the discussion.

The main problem with this study is that, if such information on Late Weichselian temperatures is published without a critical selection of data and clear indications of uncertainty limits, readers without much knowledge of the field in question and a detailed information on background data, may take statements and numbers indicated on maps and in tables as valid proxy data. Unfortunately, this is not the case for several of the borehole data applied here. This is a pity, since the topic of this study is clearly of significant interest.

---

## Referee Comment (RC3) · Anonymous Referee #3 · 1 Sep 2019

This study seeks to use borehole paleo-temperatures – whereby measured borehole temperature profiles are inverted to yield surface temperature histories – to infer whether or not certain boreholes were beneath the Scandinavian Ice Sheet during the Late Weichselian (25 to 12 kaBP). While the concept is interesting, the study has two significant issues. Firstly, the inversion methodology is far from a "gold standard" approach and is not sufficiently described to be reproducible. Secondly, the interpretation and discussion of the Scandinavian Ice Sheet extent and thickness is entirely disconnected from increasingly reliable numerical simulations of paleo ice-sheet configuration; important discrepancies and alternative explanations from such simulations are ignored. For these reasons, the usefulness of the present study is restricted to the point that it will likely negatively impact the journal's citation rate.

[Figure]

Page 1 Line 30: Arguing against peer-reviewed published studies with non-peer-reviewed conference proceedings is not good practice.

Page 2 Line 4: Modelling Insight: In light of substantial numerical modelling efforts, it is no longer acceptable to argue that the Scandinavian Ice Sheet was actually "scattered glacial domes". All available evidence suggests that the Scandinavian Ice Sheet was contiguous. See Nu et al. (2019: https://doi.org/10.1017/jog.2019.42) for the most recent PMIP simulations of the Scandinavian Ice Sheet since Last Glacial Maximum.

Page 2 Line 20: Methodological Concern: For any chance of reproducibility, the original borehole temperature profiles should also be shown, in addition to the derived surface temperature time series, for each site.

Page 2 Line 20: Methodological Concern: Given that borehole inversion is an ill-posed problem, whereby an infinite number of surface temperature histories can result in the observed borehole temperature profile, most studies now adopt a Monte Carlo approach to provide uncertainty envelopes on surface temperature histories (See: Muto et al, 2011; https://doi.org/10.1029/2011GL048086). Additionally, in this study, the "mean" profile is being taken at Outokumpu (Page 3 Line 13), while the "median" profile is being presented at Olkiluoto (Page 3 Line 20). These are not the same inversion product of a borehole temperature profile. More broadly, it seems that different inversion methods have been applied to each site.

Page 3 Line 4: Modelling Insight: It can be problematic to entirely attribute anomalously low geothermal flux – relative to the regional mean geothermal flux – to inter-glacial climate change. Significantly spatial variability in geothermal flux beneath the Scandinavian Ice Sheet has been described by other mechanisms in models (See: Naslund et al., 2005; https://doi.org/10.3189/172756405781813582). For example, the local topographic corrections to geothermal flux can be important in ice-sheet settings (See: van der Veen et al., 2007; https://doi.org/10.1029/2007GL030046).

Page 4 Line 5: Methodological Concern: I am confused how a 1000 m deep borehole

at Forsmark, can be used to reconstruct surface temperature history back to 85 kaBP in Figure 2. With most reasonable assumptions of thermal diffusivity, the deepest borehole temperatures should respond on a much shorter time-scale, and thus reflect more recent temperatures. I have admittedly not done detailed calculations myself, but the graph presented does not convince me that a diffusive temperature waves takes more than 10 ka to propagate 1000 m.

Page 4 Line 23: It is not immediately clear how surface temperatures of -8 to -18C must infer that no ice sheet was present at the borehole location, when such basal ice temperatures are found within the Greenland ice sheet today (MacGregor et al., 2016; https://doi.org/10.1002/2015JF003803). It is also very speculative to discuss presence or absence of meltwater at the base of the Scandinavian Ice Sheet – as well as its influence on ice flow – in the absence of a thermodynamic ice flow model.

Page 5 Line 25: Modelling Insight: This results interpretation seems to assume that every ground-point beneath the Scandinavian Ice Sheet only had one temperature value during the Last Glacial Period. Modelling suggests that ice-sheet may have limit cycles, whereby they thicken and warm, then flow fasters, thin and cool, and then start to thicken and warm again. This means that basal ice temperatures can flicker between warm and cold conditions. Payne, 1995 (https://doi.org/10.1029/94JB02778) mentions the Scandinavian Ice Sheet.

Page 5 Line 30: Methodological Concern: The inversions are consistently described as inferred "surface heat flux (SHF)", but in practice the derived variable is surface temperature. Precise terminology is important here, as a flux – in J/s – is a type II (prescribed flux) boundary condition while a temperature – in K – is a type I (prescribed state) boundary condition. It is unclear whether Type I or II inversion models are being applied at each borehole location.

Page 6 Line 14: The discussion of "climate sensitivity" as a parameter – "that determines how much of the additional energy incoming to the upper boundary of the

atmosphere due to the variations of the Earth's orbital parameters was finally spent to change the ground surface temperature" – seems steeped in self-citation. I am personally unaware of this parameter being widely adopted as a useful paleo climate index, but if it has been, it should be so demonstrated as being adopted beyond the author group.

Table 2: Methodological Concern: It is unclear how this modelled "amplitude of Pleistocene/Holocene warming" – which is generally approximately 20C across all sites – relates to the <10C temperature changes depicted in Figure 2. Similarly, the graphical depiction of these isotherms in Figure 3 seems to imply that Norway and Sweden have warmed in excess of 24C since the Last Glacial Period. This is significantly warmer than most previously published reconstructions.
* * *

---

## Author Comment (AC1) · 1 Nov 2019

Referee #1

We are grateful to Referee #1 for the high opinion of our paper.

RC#1: 1. It is Udryn not Udrin.

2. I propose to give a range of TLW in Table 2 for Udryn including earlier modeling result TLW=-10C of Safanda et al. 2004 (cited in the references) versus existing much later estimate by Rath and Mottaghy (2007), (-8C). I would give a range -8 to -10C as shown in both publications which used different methodology.

Author's changes: We'll take into account all the comments in the revised version.

---

## Author Comment (AC2) · 1 Nov 2019

Referee #2

AC: We are grateful to Referee #2 for his response to our paper and helpful suggestions. We appreciate the constructive feedback. We have tried to answer your questions in detail and incorporate most of your suggestions into our revised paper. Referee #2 brought up an important issue concerning not only our paper but also the overall problem of paleoclimate reconstruction from borehole temperature data. Indeed, there are some factors disturbing paleoclimate signal in temperature-depth profiles such as drilling process, ground-water movements, subsurface heterogeneities etc. It is not always possible to take into account all these factors properly. The most widely used

way to suppress the influence of non-climatic factors is the regularization of the inversion procedure. It allows suppressing spurious non-climatic anomalies but leads to the decrease of climate signal amplitude. The choice of an inversion procedure and a regularization measure are unique in each individual case. Authors of the cited papers commonly analyze different approaches and discuss in details the obtained results. However, the use of different inversion techniques makes it difficult the comparing and integrating of the reconstructed GST histories. Under the circumstances, we have accepted the following methodology. 1. If the cited paper deals with several versions of the GST reconstruction obtained by different techniques and regularization approaches we choose that one which demonstrates the minimal suppression and maximal amplitude of the reconstructed GST history. A large difference of the obtained LW temperatures and their apparent spatial correlation suggest a slight impact of non-climatic factors. 2. We did not consider an entire reconstructed temperature history, but only the Late Weichselian temperatures. The last glaciation left the most noticeable trace in the present subsurface thermal field. This trace has not been affected by the previous climate history. Temperature anomaly caused by the last glaciation extends from a depth of approximately 500 m to 1.5 – 2 km. The GST reconstructions of the last glacial period are disturbed less (versus the reconstructions of more recent climate events) by local subsurface heterogeneities and ground-water movements. Therefore, the LW temperature estimates we consider seem to be quite reliable.

Author's changes: We will significantly extend the Data section and rename it as Data and Methodology. In addition, we will add LW temperature values obtained by different techniques as well as error bars and limits of the temperature history's averaging intervals in Table 1.

Specific comments

RC#2: Still, the main problem here is the quality of the applied GST histories. The data from Kukkonen et al. (2011) from the Outokumpu . . . seem of good quality. Most other results are of far less quality. This is apparent from Fig. 2, which shows the applied

GST reconstructions. Starting from the top, with SG-3, we see an almost linear trend of increasing temperature (from c. -3_C to positive) from between 10 and 5 kyr BP and up to 1 kyr BP (present GST is +1.5_C). Similar unrealistic long trends of temperature increase up to recent times are seen also in the data from Forsmark, Laxemar and Ullrik. These 'reconstructions' are clearly inconsistent with the general knowledge of past climate in these areas as well as inconsistent with the applied data from boreholes of better quality.

AC: All the applied GST reconstructions and other estimates were published in peer-reviewed journals. The applied inversion methods were described in detail and the validity of the obtained results was discussed in these papers. Six of eleven reconstructions were obtained by V. Rath who nowadays is an acknowledge leader in the field of paleoclimate interpretation of borehole temperature data. The Technical Report was published after the publication of the discussion version of our paper. It details paleoclimate interpretation of the data obtained from the Forsmark and Laxemar boreholes on 63 pages - Rath V, Sundberg J, Näslund JO, Liljedahl LC. Paleoclimatic inversion of temperature profiles from deep boreholes at Forsmark and Laxemar. Technical Report TR-18-06, April 2019 (https://www.skb.com/publication/ 2493035/).

Author's changes: We will add the reference on this Report to the revised paper.

AC: Referee #2 means "consistence with the general knowledge of past climate in these areas" by an "apparent" quality criterion of the reconstructions. We cannot completely agree with such view point. Ground surface temperature history might be significantly more complex than surface air temperature history. It is caused not only by climate change but also by the characteristics of vegetation, snow cover as well as the presence of water at the surface and ice sheets. Thus, the GST reconstructions from the Forsmark and Laxemar boreholes show long history of flooding and draining of the region after deglaciation (see the Technical Report mentioned above). Perhaps, the SG-3 and Ullrigg location areas were also influenced by flooding and following draining due to the isostatic uplift. However, our plans did not include the analysis of postglacial

history of the region. Perhaps, the applied reconstructions do not have the required quality for such an analysis. We set a goal to estimate the thermal regime in the Late Weichselian and to analyze spatial distribution of the reconstructed LW temperatures by the way of their comparing with independent evidences including geological and seismological ones. RC#2: Among the sites with very low LW GST estimates are the Kola (C-1), Krl and Onega boreholes. For Onega, we see an 'unexpected' drop in temperatures by more than 5_C from c. 5 to 2.5 kyr BP. Looking into details of original borehole temperature data (in Demezhko et al. 2013), we observe too high near surface temperatures (c. 11_C significantly above present day GST of c. 5.5_C). The applied borehole temperature data are clearly disturbed by the drilling process. A correction is attempted resulting in an unrealistic 'warming period' and the above 'unrealistic' temperature drop. This results in too large amplitude of the temperature rise from c. 20 to 10 kyr BP and significant uncertainty on the Late Weichselian temperature estimate of - 14.5_C. AC: Perhaps, 'unexpected drop' from 5 to 2.5 kyr BP for the reconstruction from the Onega borehole might be associated with unaccounted influence of non-climatic factors. However, we are interested in much earlier times that is the Late Weichselian. The GST estimate of -14.5 C is proved by the results of mathematical simulations for Karelia (Forsström P.-L. Through a glacial cycle: simulation of the Eurasian ice sheet dynamics during the last glaciation. Annales Academiae Scientiarum Fennicae, Geologica-Geographica. 2005, 168, 94 pp.). The reference to (Forsström, 2005) is given in the cited paper (Demezhko et al., 2013).

Author's changes: We will include the corresponding clarification in the revised text.

RC#2: For the Kola (C-1) site, a LW GST estimate of -18_C is indicated. The problem here is a 'deep narrow cooling' between c. 35 and 20 kyr BP.

Author's changes: We have defined more exactly this LW temperature value. Now it is a mean temperature at the GST minimum over the period 30-20 kyr BP equal to -16.8 °C.

RC#2: For borehole Krl, no GST history is given, and the low LW value of -15°C is obtain by 'selecting' an estimated 'unperturbed' heat flow of 40 mW/m2 without any mention of modelling procedure, nor information on deep background heat flow (why 40?).

AC: The paper (Kukkonen et al., 1998) does not provide a direct estimate of LW temperature value. The forward modelling suggests that the very low temperature gradients measured in this area "could be attributed to very low ground temperatures (-10 to -15°C) during the glaciation". LW temperature of -10 °C corresponds to a heat flow value of 19 to 32 mW/m2 while a heat flow value of 26 to 40 mW/m2 could be attributed to LW GST value of -15 °C. Later heat flow estimates for this region made using a large number of boreholes taking into account paleoclimate impact (Majorowicz, J., & Wybraniec, S. 2011. New terrestrial heat flow map of Europe after regional paleoclimatic correction application. International Journal of Earth Sciences, 100(4), 881-887) are equal to 40-50 mW/m2. Against this background we have chosen lower LW temperature value (-15 °C).

Author's changes: We will clarify our choice in the revised paper.

RC#2: The 'mathematical/numerical' contouring of data in Fig. 3 and the extrapolation into the Scandinavian region (with "a center of warming" located in the North Atlantic) has so large uncertainties in the region of study, that it should not be used for a detailed treatment as in Table 2 and Fig. 4 and associated discussion.

AC: The mentioned 'mathematical/numerical' model was widely discussed during the publication process in "Climate of the Past". Its statistical robustness was proved by the bootstrap analysis method. The model provides the least reliable estimates for the northwest parts of Norway and Sweden because there is no reference reconstructions here. However, the applied GST reconstructions are located outside of this region. The modelled GST estimates are in a good agreement with the obtained LW temperatures for the regions where the ice sheet existed not long and contrast strongly with those

obtained in the regions covered by the ice sheet most of the Late Weichselian.

Author's changes: We will limit the isoanomalies to the value of $\Delta T=20ÐŽ$ in the corrected text of the paper. To illustrate how the GST estimation uncertainties affect our conclusions we will add the GST values obtained using different methods as well as uncertainty envelopes in Table 1 and figures.

RC#2: The notion of a potential correlation between the region of very low modern seismicity (Fig. 6) and very low LW temperatures seems highly speculative. Most of Finland has very little seismicity, also in areas of significant ice thickness towards Gulf of Bothnia. The highest current seismicity is in southwester Norway in areas along the ice sheet margin.

AC: In Fig. 6 the territory of Finland is marked as the area with the moderate seismicity while there is no seismicity at all on the eastern and southern margins of the studied region with the lowest LW temperatures (blue circles). We mentioned in the paper that one can expect the existence of correlation (a weak one in general) between LW temperatures and seismicity caused by isostatic uplift and not by tectonics.

---

## Author Comment (AC3) · 1 Nov 2019

Referee #3

AC: Our paper is aimed at the evaluation how the existing Late Weichselian GST reconstructions obtained from borehole data using different approaches are comparable each other and with the available climate evidences in the studied region. Because nowadays there is no valid proxies of the thermal state at the base of the Pleistocene ice sheets, borehole GST estimates might be of interest whether for multiproxy data analysis or as a source of input data for mathematical simulation of ice sheets.

We are grateful to Referee #3 for his helpful critical comments. We appreciate the constructive feedback. We agree that it is necessary to include the data on the uncertainties of the applied GST estimates to clarify the validity of the made conclusions.

Author's changes: We will add in the revised text an additional information as well as an explanation of methodology that will help to get an idea of the reliability of the data.

General comments

Referee #3 has brought up two significant issues:

Firstly, the inversion methodology is far from a "gold standard" approach and is not sufficiently described to be reproducible. And hereinafter: …most studies now adopt a Monte Carlo approach to provide uncertainty envelopes on surface temperature histories (See: Muto et al, 2011).

AC: Since the GST reconstructing using borehole temperature data is based on the mathematical simulation of the current temperature-depth distribution in subsurface (i.e. an inversion procedure), there is not and cannot be a "gold standard" in this case. Like there is not a "gold standard" in the modeling of climate or an ice sheet dynamics. All approaches used for estimation of GST histories from borehole data and their validity were described and published in peer-reviewed journals (there are corresponding references in the text). Monte Carlo approach is one of such methods. However, it is not applied quite often.

Concerning "to provide uncertainty envelopes". The main sources of uncertainties in geothermal method are connected with the uncertainty of a stationary model of temperature distribution caused by subsurface heterogeneities, ground-water movement, insufficient restoration of thermal regime after the drilling completion etc. The uncertainty envelopes given for the reconstructions are usually reproduce methodological uncertainties caused by the GST fitting features using the chosen method under the accepted stationary model.

Author's changes: To illustrate the validity of our conclusions we will supplement Table 1 with the LW temperature estimates and uncertainty envelopes obtained by different

authors using different inversion methods (including Monte Carlo approach). All the additional estimates and limits of uncertainties will be used while analyzing spatial features of the LW GST values (Fig. 4). We will show the GST history averaging intervals and limits of uncertainties on the diagram of GST reconstructions.

RC#3: Secondly, the interpretation and discussion of the Scandinavian Ice Sheet extent and thickness is entirely disconnected from increasingly reliable numerical simulations of paleo ice-sheet configuration; important discrepancies and alternative explanations from such simulations are ignored.

AC: We agree that the methods of numerical simulations of paleo ice-sheets have consistently been improved. However, our plans do not include a detailed comparison of the GST reconstructions with the results of simulations. We contrarily believe that the use of independent GST geothermal estimates as the input data in numerical simulation of ice sheets could allow improving the reliability of simulation results. Our results are in a good agreement with the reconstructions of Hughes et al., (2016) based on proxy data and simulation results.

Specific comments

RC#3: Page 1 Line 30: Arguing against peer-reviewed published studies with non-peerreviewed conference proceedings is not good practice

Page 2 Line 4: Modelling Insight: In light of substantial numerical modelling efforts, it is no longer acceptable to argue that the Scandinavian Ice Sheet was actually "scattered glacial domes". All available evidence suggests that the Scandinavian Ice Sheet was contiguous. See Nu et al. (2019: https://doi.org/10.1017/jog.2019.42) for the most recent PMIP simulations of the Scandinavian Ice Sheet since Last Glacial Maximum.

AC: The reference to the Conference Proceedings is included only to demonstrate the existence of different viewpoints on the Scandinavian Ice Sheet extent. We did not plan to analyze in details these viewpoints. Our conclusions do not support the

separate ice domes paradigm (at least, within the spatial and temporal resolution power of geothermal method).

RC#3: Page 2 Line 20: Methodological Concern: For any chance of reproducibility, the original borehole temperature profiles should also be shown, in addition to the derived surface temperature time series, for each site.

Author's changes: We will include the figure showing measured borehole temperature profiles in the revised text. It is necessary to note that measured temperature profiles reproduce not only the influence of past temperature changes but also the impact of non-climatic factors such as subsurface heterogeneities, insufficient restoration of thermal regime after drilling completion etc. The methodology for taking into account these factors is described in the cited papers.

RC#3: Page 2 Line 20: Methodological Concern: Given that borehole inversion is an ill-posed problem, whereby an infinite number of surface temperature histories can result in the observed borehole temperature profile, most studies now adopt a Monte Carlo approach to provide uncertainty envelopes on surface temperature histories (See: Muto et al, 2011; https://doi.org/10.1029/2011GL048086). Additionally, in this study, the "mean" profile is being taken at Outokumpu (Page 3 Line 13), while the "median" profile is being presented at Olkiluoto (Page 3 Line 20). These are not the same inversion product of a borehole temperature profile. More broadly, it seems that different inversion methods have been applied to each site.

AC: Definitely. Unfortunately, the existed geothermal estimates of the GST histories were obtained using different techniques. The choice of the technique depended not only on individual preferences of the author but also on available information on thermal properties of rocks, the duration of thermal regime restoration etc. Perhaps, it will be possible to reconstruct paleoclimate from most of the available temperature-depth profiles within a common approach in the future. However, the aim of our work was to demonstrate that even the use of various techniques allow obtaining a spatially correlated result.

Author's changes: We will add the corresponding clarifications in the description of the paper's goal. In addition, we will add LW temperature values obtained by different techniques as well as error bars and limits of the temperature history's averaging intervals in Table 1 and figures.

RC#3: Page 3 Line 4: Modelling Insight: It can be problematic to entirely attribute anomalously low geothermal flux – relative to the regional mean geothermal flux – to inter-glacial climate change. Significantly spatial variability in geothermal flux beneath the Scandinavian Ice Sheet has been described by other mechanisms in models (See: Naslund et al., 2005; https://doi.org/10.3189/172756405781813582). For example, the local topographic corrections to geothermal flux can be important in ice-sheet settings (See: van der Veen et al., 2007; https://doi.org/10.1029/2007GL030046).

AC: The direct estimate of LW temperature value is not presented in (Kukkonen et al., 1998). The forward models suggest that the very low temperature gradients measured in this area "could be attributed to very low ground temperatures (-10 to -15°C) during the glaciation". LW temperature of -10 °C corresponds to a heat flow value of 19 to 32 mW/m2 while a heat flow value of 26 to 40 mW/m2 could be attributed to LW GST value of -15 °C. Later heat flow estimates for this region made using a large number of boreholes taking into account paleoclimate impact (Majorowicz, J., & Wybraniec, S. 2011. New terrestrial heat flow map of Europe after regional paleoclimatic correction application. International Journal of Earth Sciences, 100(4), 881-887) are equal to 40-50 mW/m2. Against this background we have chosen lower LW temperature value (-15 °C).

Author's changes: We will clarify our choice in the revised paper.

AC: In (Naslund et al., 2005) mentioned by Referee #3 high-resolution heat flow estimates were calculated only for Sweden and Finland with an average value of 49 mW m–2 (it is even higher than the estimate we used – 40 mW m–2). For Karelia (Naslund

et al., 2005) provides unadjusted data by H.N. Pollack and others. Due to the poorly broken relief near the Krl borehole the correction for topography of heat flow estimates is not required here.

RC#3: Page 4 Line 5: Methodological Concern: I am confused how a 1000 m deep borehole C2 at Forsmark, can be used to reconstruct surface temperature history back to 85 kaBP in Figure 2. With most reasonable assumptions of thermal diffusivity, the deepest borehole temperatures should respond on a much shorter time-scale, and thus reflect more recent temperatures. I have admittedly not done detailed calculations myself, but the graph presented does not convince me that a diffusive temperature waves takes more than 10 ka to propagate 1000 m.

AC: Certainly, the 1000-m temperature-depth profile by itself cannot provide the GST reconstruction for the past 85 kyr. The Technical Report (Rath et al., 2019) provides a set of GST histories that are significantly different for t > 10 kyr ago depending on the heat flow value HFD. For the Late Weichselian GST estimate varies from -2 C under HFD of 60 mW/m2 to +1.5 C under HFD of 48 mW/m2 (see fig. 4-1 in (Rath V, Sundberg J, Näslund JO, Liljedahl LC. Paleoclimatic inversion of temperature profiles from deep boreholes at Forsmark and Laxemar. Technical Report TR-18-06, April 2019. https://www.skb.com/publication/ 2493035/)). This Technical Report was published after the publication of our paper in CPD.

Author's changes: We will add the reference on this Report as well as both LW GST values to the revised paper.

RC#3: Page 4 Line 23: It is not immediately clear how surface temperatures of -8 to -18C must infer that no ice sheet was present at the borehole location, when such basal ice temperatures are found within the Greenland ice sheet today (MacGregor et al., 2016; https://doi.org/10.1002/2015JF003803). It is also very speculative to discuss presence or absence of meltwater at the base of the Scandinavian Ice Sheet – as well as its influence on ice flow – in the absence of a thermodynamic ice flow model.

AC: We wrote: "Such temperature regime at the ground surface points to the existence of ice-free conditions during much of the Late Weichselian. At least if the glacier existed here then it was not for a long time and its thickness was not so significant to make any noticeable contribution in the modern thermal field". In modern Greenland low GSTs are observed both on its ice-free margins (especially in the North) and in the middle of the ice sheet. In addition, we explained low temperatures in the central parts of ice sheets (Page 8 Line 23): "The simulation results (Demezhko et al., 2007) showed that temperature at the base of the glacier depends on the balance of heat flow, vertical advection and a glacier height influences. Under significant glacier height and high vertical advection rate low temperatures from a glacier surface are transmitted to its basement. Convective heat transfer mechanism is more effective than conductive one. As a result, the glacial basement may cools". To our mind, all the answers are already given in the text of the paper. Concerning the discussion about "presence or absence of meltwater at the base of the Scandinavian Ice Sheet". Water phase state is determined by temperature, pressure and the existence of impurities. All these factors are mentioned in the paper. To better illustrate the phase state, we will add the ice/water phase state boundaries at several values of an ice sheet height to Figure 4.

RC#3: Page 5 Line 25: Modelling Insight: This results interpretation seems to assume that every ground-point beneath the Scandinavian Ice Sheet only had one temperature value during the Last Glacial Period. Modelling suggests that ice-sheet may have limit cycles, whereby they thicken and warm, then flow fasters, thin and cool, and then start to thicken and warm again. This means that basal ice temperatures can flicker between warm and cold conditions. Payne, 1995 (https://doi.org/10.1029/94JB02778) mentions the Scandinavian Ice Sheet.

AC: Geothermal reconstructions of the mean LW GST leave open a possibility of significant short-time variations of temperature within the averaging time. We wrote (Page 7 Line 26) "The inversion of borehole temperature data allows estimating the ground surface temperature not at specific time t in the past but the mean temperature for the

period t±t/3 (Demezhko and Shchapov, 2001). For example, temperature 21 kyr BP on the reconstructed GST history represents an average value over a period of 14 – 28 kyr BP".

RC#3: Page 5 Line 30: Methodological Concern: The inversions are consistently described as inferred surface heat flux (SHF), but in practice the derived variable is surface temperature. Precise terminology is important here, as a flux – in J/s – is a type II (prescribed flux) boundary condition while a temperature – in K – is a type I (prescribed state) boundary condition. It is unclear whether Type I or II inversion models are being applied at each borehole location.

AC: There exist two ways of the surface heat flux evaluation. The first one is the reconstruction of the SHF history directly from temperature-depth profile using the inversion procedure (type I boundary condition; Beltrami H. Surface Heat Flux Histories from Geothermal Data: Inferences from Inversion / Geophys. Res. Lett., 2001, 28(4), p. 655-658). The second way (that we used) is to transform GST history obtained for the type I boundary conditions (Beltrami, 2002; Huang, 2006; Demezhko et al., 2013; Demezhko, Gornostaeva, 2014; 2015 a,b).

Author's changes: We will clarify this issue in Section 4 - Distribution of the ground surface heat fluxes.

RC#3: Page 6 Line 14: The discussion of "climate sensitivity" as a parameter – "that determines how much of the additional energy incoming to the upper boundary of the atmosphere due to the variations of the Earth's orbital parameters was finally spent to change the ground surface temperature" – seems steeped in self-citation. I am personally unaware of this parameter being widely adopted as a useful paleo climate index, but if it has been, it should be so demonstrated as being adopted beyond the author group.

AC: Indeed, we have proposed the parameter of climate sensitivity not so long ago as an alternative to the traditional one representing temperature reaction to changes

of incoming radiation. It is natural that this parameter does not become a frequent practice yet. We suppose that this is not a reason not to mention the climate sensitivity in the frame of our paper and to compare the estimates of climate sensitivity made in Fennoscandia with those obtained in ice-free regions.

RC#3: Table 2: Methodological Concern: It is unclear how this modelled "amplitude of Pleistocene/Holocene warming" – which is generally approximately 20C across all sites –relates to the <10C temperature changes depicted in Figure 2. Similarly, the graphical depiction of these isotherms in Figure 3 seems to imply that Norway and Sweden have warmed in excess of 24C since the Last Glacial Period. This is significantly warmer than most previously published reconstructions.

AC: That is right. The inconsistencies between the applied reconstructions and the empirical model data reveal the warming effect of the ice sheet. On the contrary, for the regions where there was no ice sheet most of the Pleistocene the GST reconstructions agree well with the model. The empirical model of spatial distribution of Pleistocene/Holocene warming amplitudes summarizes a number of long-term geothermal reconstructions obtained earlier in North Eurasia (Demezhko et al., 2007). Generally, these deltaT estimates were obtained outside the studied region on the territories free of the Pleistocene ice sheets. In the paper we compare not amplitudes but LW temperatures – Page 5 Line 19: "Subtracting the modeling amplitudes of Pleistocene/Holocene warming from the current mean annual ground surface temperatures $T_c$ at the borehole location points, we obtain rough estimates of the 'normal' GST TLW-mod, which characterize the temperature regime while there was no ice sheet (Tab. 2)". This comparison reveals the existence of two separate clusters (Fig. 4). According to the reconstruction made by (Dahl-Jensen, D. et al. Past temperature directly from the Greenland ice sheet, Science, 282, 268–271, 1998) an amplitude of the Pleistocene/Holocene warming in Greenland that is located equivalently far from a hypothetical warming center like Norway or so is equal to 23 K. However, for the northwest parts of Norway and Sweden the model gives very unreliable estimates of the amplitude because there is

no reference reconstructions here.

Author's changes: We will limit the isoanomalies in Fig. 3 to the value of 20K in the revised text of the paper.